# Botulinum toxin in the treatment of partially accommodative esotropia with high AC/A ratio

Jaime Tejedor[1,2]*, Francisco J. Gutiérrez-Carmona[1,3]

**1** Department of Ophthalmology, Hospital Ramón y Cajal, Madrid, Spain, **2** Department of Neuroscience, Universidad Autónoma de Madrid, Madrid, Spain, **3** Universidad Alfonso X El Sabio, Madrid, Spain

* jaime.tejedor@telefonica.net

## Abstract

### Purpose

To study the outcome of botulinum toxin (BTX) treatment (group 1) in partially accommodative esotropia with high accommodative convergence/accommodation (AC/A) ratio, in comparison with bilateral medial rectus muscles recessions and posterior fixation (group 2).

### Methods

In a retrospective comparative study, children aged 3–8 years old treated between 2011 and 2016, with partially accommodative esotropia with high AC/A ratio, deviation at distance of 10 prism diopters or more, and at least 1 year of follow-up, were included. Visual acuity, alternate prism and cover test, stereoacuity, biomicroscopy, and cycloplegic retinoscopy were carried out at initial, baseline visit, 6 months and 1 year after BTX injection or surgery. Main outcome variables were deviation at distance and near, improvement in stereoacuity, and percentage of success. We used multiple regression or proportional odds analysis to control for potential confounding variables.

### Results

Of 95 patients, 84 were eligible, 48 children in group 1 and 36 in group 2. Deviation and stereoacuity were similar in the two groups at 6 months, but significantly better in the BTX group at 1 year (median distance deviation 0 prism diopters vs 5 prism diopters, p<0.01), although differences were not clinically relevant. Percentage of success was also significantly better only at 1 year (93% vs 72%, p = 0.01). Change in distance-near disparity was not significantly different in the two groups in the period of study.

### Conclusions

Botulinum toxin could be superior to, or as effective as surgery, at middle term, in the treatment of partially accommodative esotropia with high AC/A ratio.

**Data Availability Statement:** All relevant data are in the paper and its Supporting Information files.

**Funding:** JT received grant MINECO UAMA13-4E-2192 The funders had no role in study design, data

collection and analysis, decision to publish, or preparation of the manuscript

**Competing interests:** The authors have declared that no competing interests exist.

## Introduction

In children with accommodative esotropia type high accommodative convergence/accommodation ratio, correction of hyperopia may control deviation at distance, with persistent deviation at near, but in some patients yet there is residual deviation at distance with glasses on (partially accommodative). When distance deviation equals or exceeds 10 prism diopters (PD), our aim is to align eyes at distance with additional procedures, several of which are available. Surgery is directed to operate for the near angle[1], for the average of near and distant angles measured with the distance correction, or added recession to standard surgical dosages.[2] Another option is to recess medial rectus muscles and add posterior fixation (fadenoperation) to facilitate control of deviation at near.[3–5] We had used the latter procedure for several years, whereas botulinum toxin injection in the medial rectus muscles was not considered a primary indication for this condition. However, we had to use botulinum toxin in some cases, in particular when parents did not agree to surgery. The results obtained using botulinum toxin were considered to be satisfactory. Consequently, we started to treat children using this therapy more frequently. In this study, our purpose was to investigate the results of botulinum toxin (BTX) (group 1) compared to bilateral medial rectus muscles recessions with posterior fixation (group 2) in treating partially accommodative esotropia with high AC/A ratio.

## Methods

We retrospectively reviewed the charts of children aged 3–8 years old with the diagnosis of partially accommodative esotropia with high AC/A ratio made between 2011 and 2016 (search in the electronic clinical record system). At a significance level of 0.05, detectable difference of 10 PD, standard deviation of 12 PD, and power of 80%, a sample size of n = 26 was required in each group. The study was approved by the Institutional UAM Ethics Committee (MINECO UAMA13-4E-2192) and adhered to the tenets of the Declaration of Helsinki. After a detailed explanation of the nature of the study, informed consent was obtained for BTX injection or surgery, and for collection of the relevant data.

Distance glasses were prescribed at initial evaluation (full cycloplegic refraction). Esodeviation measured with distance correction after 2 months of wearing glasses full time (baseline visit), was used for eligibility. Children with residual esotropia were re-refracted before additional intervention was indicated. Two children in the surgery group, and three in the botulinum group required new full plus glasses prescription for 2 months before a decision was taken. Children who had esotropia at distance of at least 10 PD, and an increase in deviation by 10 PD or more for near as compared to distance, and underwent surgery by recession of medial rectus muscles and posterior fixation suture (surgery) or botulinum toxin injection in medial rectus muscles (BTX) were eligible. AC/A ratio was determined using a lens gradient method with +3.00 D lenses at near (the difference between deviation without and with the +3.00 D lens was divided by 3 to obtain the AC/A ratio; AC/A ratio > 5 was considered high). For inclusion, at least 1 year of follow up after intervention (surgery or BTX) was required. After explanation of the nature, side effects, and characteristics of the two procedures, we offered the family the option of botulinum toxin injection or surgery, and the procedure was decided in agreement with parents. Bifocal add was contemplated only after intervention when deviation at near persisted. Smallest power of bifocals required for orthotropia at near was prescribed. Parents were informed that bifocals reduced or eliminated deviation at near, but that existing data didn't show clear benefit in bifocal wearing, or at least its use was controversial.

Children with the diagnosis of eye disease, previous eye surgery, myopia greater than -0.50 D, or follow-up of less than 1 year after intervention were excluded from the analysis.

Clinical evaluation included measurement of visual acuity (logMAR HOTV or electronic ETDRS, depending on chidren's ability to cooperate; a level was considered passed when at least three of four letters were identified correctly), stereoacuity at near (Randot preschool or Randot stereotests–circles with random dot ground, Stereo Optical Inc, Chicago), deviation angle at distance (6 m) and near (35 cm) by alternate prism and cover test, and fusion (Bagolini, Worth four dot). Cycloplegic retinoscopy, as well as anterior and posterior segment routine examination were also part of the study in each follow-up visit. Stereoacuity was tranformed to log arc seconds, and prism deviation to degrees, for statistical analysis. To enable calculation of changes in stereoacuity, if the patient had no measurable stereoacuity, the next log level above the largest disparity for the test was assigned (in 0.3 log arcsec progression, 6000 arcsec, i.e. 3.78 log arcsec). This is a commonly used strategy in analysis of stereoacuity data.[6,7] We defined amblyopia as a difference of at least 2 logMAR lines in visual acuity between the two eyes. Patching (2 to 6 hours per day depending on clinician's judgement) was used for the treatment of amblyopia.

We injected 2.5 to 3 IU of Botulinum toxin in both medial rectus muscles (group 1) or did recession of both medial rectus muscles for an average deviation between distance and near (AAO surgical guidelines for esodeviation[8]) with posterior fixation (group 2). Recessions of medial rectus muscles ranged between 3 and 4.5 mm. Posterior fixation was done at 13–14 mm from the muscle insertion, at approximately the location of the muscle pulley. Although this was identified during surgery, we nevertheless chose to use a classical scleral fixation technique to secure the muscle, with a nonabsorbale 5–0 Dacron suture, and not a modified pulley fixation technique, as described by Clark et al.[5]

We employed linear multiple regression, when the distribution of variables was compatible with normality, or proportional odds, when the distribution was not normal, to control for potential confounders. Preintervention variables included in the stepwise multivariate analysis (we controlled for) were age, sex, refraction, baseline deviation (at distance and near with distance correction), lines of difference in visual acuity between the two eyes, and stereoacuity. We also used unpaired t test (with Bonferroni correction to manage potential error for multiple comparisons) and Chi-square test. Values obtained in the analysis, were reconverted to the original scale, for a more familiar notation to the clinician. Graphical representation of variable distribution and Shapiro-Wilk test were used to test for normality.

The primary outcome variable was deviation at distance and near, change in distance-near disparity, and success (defined as deviation at distance $\leq$ 8 DP and fusion at near) at 6 months and 1 year. A secondary outcome variable was improvement in stereoacuity.

## Results

Of 95 children, 84 were eligible, and 11 were excluded due to short follow-up (2), insufficient data (6), or myopia (3). Power of the statistical analyses done in this study was at least 80%. All patients wore glasses when hyperopic of $\geq$ 1.50 D, until evaluation 2 months after initial visit (2-month visit was considered baseline visit). Baseline characteristics of children included are summarized in Table 1. We did not find differences between the two treatment groups except for refraction, which was slightly larger in the botulinum group.

Botulinum toxin injection or surgery was indicated, after explanation of the nature, advantages and disadvantages of the procedures, when deviation at distance of at least 10 PD, and at least 10 PD greater at near, was observed in the baseline visit. We were in favor of doing surgery but when parents were reluctant, or taking into consideration that surgery required more time of general anesthesia and was more invasive, botulinum toxin injection was chosen. Amblyopia was treated, when present, for no more than 4 months before intervention (botulinum or surgery), so intervention was not delayed.

**Table 1. Characteristics of patients included[†].**

| Preintervention variables | BTX (n = 48) | Surgery (n = 36) | p |
|---|---|---|---|
| Age (years) | 5 (3–8) | 5 (3–8) | 0.9 |
| Refraction (diopters) | 3 (2.25–5) | 2.75 (2.25–3.25) | <0.01 |
| Distance deviation (deg) | 6.8 (5.7–11.3) | 8.5 (5.7–13.5) | 0.07 |
| Near deviation (deg) | 15.6 (11.3–16.7) | 15.6 (13.5–19.8) | 0.4 |
| logMAR lines of difference | 0 (0–3) | 0 (0–3) | 0.3 |
| Stereoacuity (log arc sec) | 2.9 (2.6–3.78) | 2.9 (2.6–3.78) | 0.7 |
| Sex (female)[‡] | 26/48 54.2 | 19/36 52.8 | 0.8 |

[†]Median (min-max) except for sex variable

[‡]proportion

Six months after Botox injection or surgery, the deviation (with distance glasses) at distance and near, as well as stereoacuity, were similar in the two groups. However, the deviations in the surgery group were significantly larger and stereoacuity in the BTX group significantly better at 1 year (see Table 2 and Figs 1 and 2). Some variable differences at 6 months were near significance (e.g., difference in distance deviation and stereoacuity). Most differences encountered at 6 or 12 months, being significant or not, were of little clinical relevance, because they were in the limits of reproducibility of the prism and alternate cover test.[7]

Change in distance-near disparity was not significantly different between the botulinum and surgery group at 6 months (median 6.6 deg [12 PD], range 0–8.8 deg [0–16 PD] and 4.8 deg [9 PD], range 0–8.8 [0–16 PD], respectively; p = 0.05) and 12 months (median 6.6 deg [12 PD] range 0–8.8 deg [0–16 PD] and 5.4 deg [10 PD], range 3.1–8.8 deg [6–16 PD] respectively; p = 0.14).

Percentage of success (defined as deviation at distance ≤ 8 DP and fusion at near with Bagolini lenses/Worth four dot) was similar at 6 months in the botulinum and surgery group (45 of 48 [93.7%], and 31 of 36 [86.1%], respectively, p = 0.23) but smaller in the surgery group at 1 year (45 of 48 [93.7%], and 26 of 36 [72.2%], respectively, p = 0.01). The odds in favor of success of the botulinum group was 2.4 times that of the surgery group at 6 months (95% CI:0–2.5), and 5.7 times at 1 year (95% CI:1.4–22.8) (see Table 3). Apparently, the slight trend

**Table 2. Motor outcome after surgery or botulinum toxin injection.**

| | Botulinum toxin | Surgery | p |
|---|---|---|---|
| Deviation at 6 months (median [range]) | | | |
| Distance | | | |
| Degrees | 0 [0–4.5] | 2.3 [0–5.7] | 0.05 |
| Prism Diopters | 0 [0–8] | 4 [0–10] | |
| Near | | | |
| Degrees | 2.3 [0–9] | 4.5 [0–10] | 0.15 |
| Prism Diopters | 4 [0–16] | 8 [0–18] | |
| Deviation at 12 months (median [range]) | | | |
| Distance | | | |
| Degrees | 0 [0–3.4] | 2.8 [0–6.8] | 0.01 |
| Prism Diopters | 0 [0–6] | 5 [0–12] | |
| Near | | | |
| Degrees | 2.2 [0–6.8] | 4.5 [0–7.9] | 0.03 |
| Prism Diopters | 4 [0–12] | 8 [0–14] | |

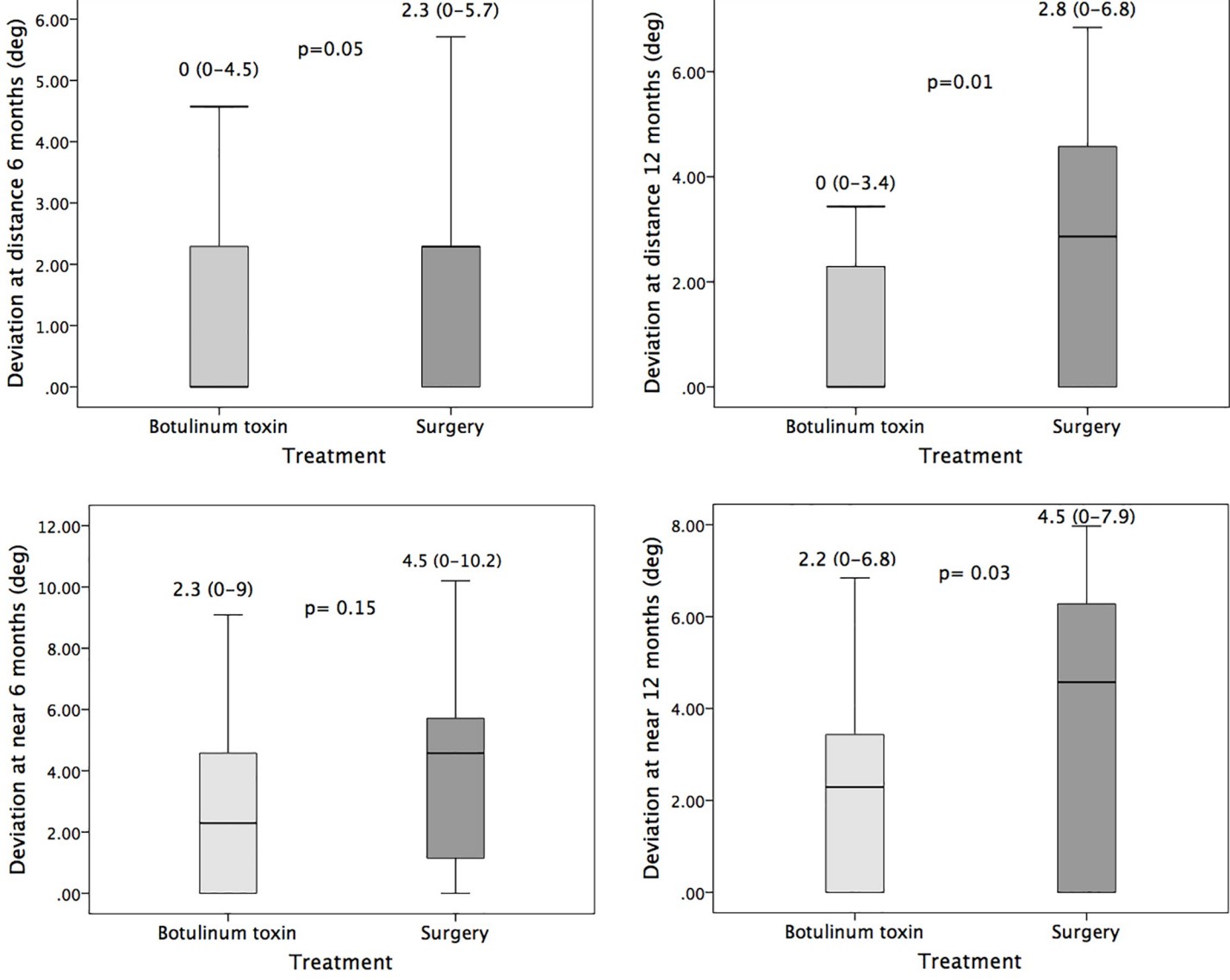

**Fig 1. Motor outcome at 6 and 12 months.** Boxplot showing deviation at distance and near (degrees) at 6 months (A,C) and 12 months (B,D) after botulinum toxin or surgical treatment. Deviation was similar in the two treatment groups group at 6 months, but was smaller in the botulinum group at 1 year, although differences were not of remarkable clinical relevance. The bottom and top of each box represent the 25th and 75th percentiles (the lower and upper quartiles), respectively; the solid line band in the box is the median (50th percentile). The upper whisker is the smaller of the 75th percentile plus 1.5 times the interquartile range (IQR) or the maximum value. The lower whisker is the larger of the 25th percentile minus 1.5 times the IQR or the minimum value.

toward recurrence of deviation was larger in the surgery group. Four patients in the botulinum group and 3 patients in the surgery group required change (additional plus) in refraction at 6 months, whereas 5 patients in the botulinum and 4 in the surgery group required new glasses prescription at one year.

Percentage of bifocal use was significantly lower in the botulinum toxin than in the surgery group (8 of 48 [17%] vs 14 of 36 [39%], respectively, p = 0.02). The odds in favor of using bifocals in the surgery group was 3 times (95% CI:1.1–8.7) that of the botulinum group (Table 3). Frequency of a new procedure during the year of study follow-up (surgery, when deviation at distance was greater than 8 PD), was similar in the two groups (3 of 48 in the botulinum group [6.2%]) vs 7 of 36 in the surgery group [19.4%], p = 0.09). Two of these children in the

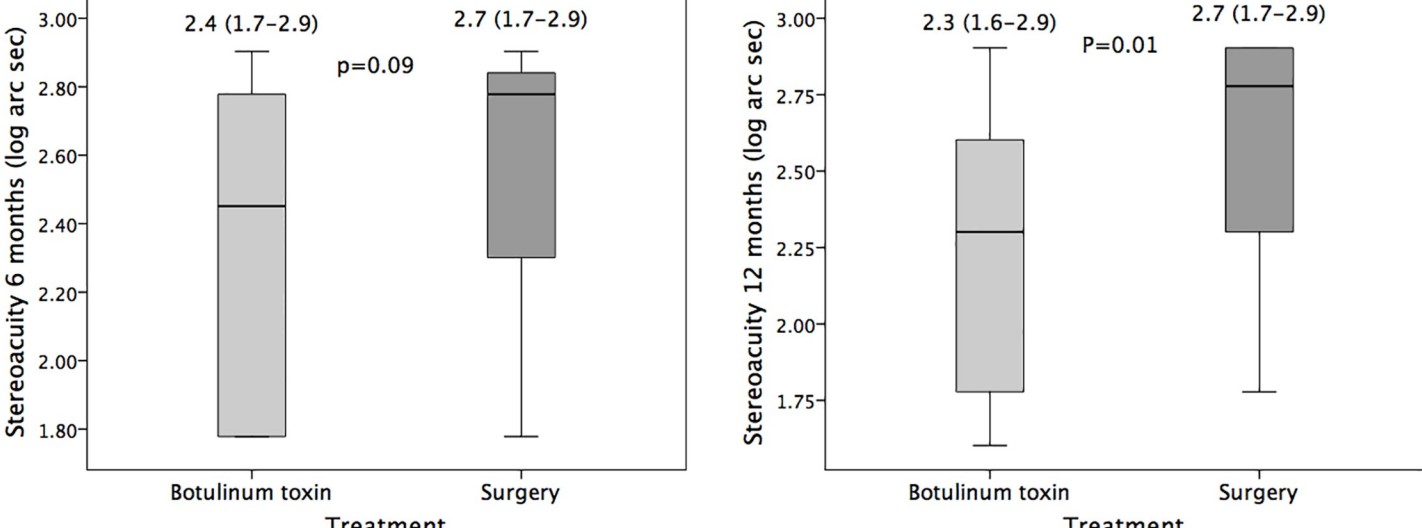

**Fig 2. Stereoacuity outcome at 6 and 12 months.** Boxplot showing stereoacuity (log arc seconds) at 6 months (A) and 12 months (B) after botulinum toxin or surgical treatment. Stereoacuity was similar in the two treatment groups group at 6 months, but was better in the botulinum group at 1 year, although differences were not of particular clinical relevance. The bottom and top of each box represent the 25th and 75th percentiles (the lower and upper quartiles), respectively; the solid line band in the box is the median (50th percentile). The upper whisker is the smaller of the 75th percentile plus 1.5 times the interquartile range (IQR) or the maximum value. The lower whisker is the larger of the 25th percentile minus 1.5 times the IQR or the minimum value.

botulinum group and 3 in the surgery group were prescribed additional hyperopic correction before indication of reintervention. Three of the 10 'unsuccessful result' patients in the surgery group were not reoperated because deviation at distance was 10 PD, but it was not apparent, and they were able to fuse at near.

We used multivariate analysis to identify significant predictors of outcome variables (see Table 4). Significant predictors of deviation and success at 12 months were treatment modality and lines of difference in visual acuity between the two eyes, respectively. A significant predictor of deviation and success at 6 months was deviation before intervention.

## Discussion

We have studied the outcome of patients with partially accommodative esotropia with high AC/A ratio who required treatment for deviation at distance and near, treated with botulinum toxin. We used patients treated by bilateral medial rectus muscles recessions and posterior fixation, during approximately the same period of time, as a control group. We did not conduct a randomized controlled trial due to the relatively small number of children with these characteristics that we treated every year, thus a multicenter study would be required.

Percentage of success, deviation and stereoacuity were not significantly different in the two groups at 6 months, although difference in some variables almost reached significance. The

**Table 3. Odds of failure and bifocal use.**

| Variable | BTX[†] | Surgery[†] | Odds ratio |
|---|---|---|---|
| Failure at 6 m | 3/48 6.2 (-0.5–13.1) | 5/36 13.9 (2.5–25.1) | 2.4 (0–2.5) |
| Failure at 12 m | 3/48 6.2 (-0.5–13.1) | 10/36 27.8 (13.1–42.4) | 5.7 (1.4–22.8) |
| Bifocal use | 8/45 16.7 (6.1–27.2) | 14/36 38.9 (22.9–54.8) | 3.1 (1.1–8.7) |

[†]Proportion (95% CI)

**Table 4. Predictors for each outcome variable.**

| Outcome Variable | Significant predictor variables | p |
|---|---|---|
| Deviation at distance 6 m (deg) | Preintervention distance deviation | 0.001 |
| Deviation at distance 12 m (deg) | Treatment | 0.001 |
| | Lines of difference | <0.001 |
| Deviation at near 6 m (deg) | Preintervention near deviation | <0.001 |
| Deviation at near 12 m (deg) | Treatment | 0.002 |
| | Lines of difference | <0.001 |
| Success at 6 m | Preintervention near deviation | <0.001 |
| Success at 12 m | Treatment | 0.013 |
| | Lines of difference | 0.006 |

differences observed became significant at 12 months, in favor of the botulinum toxin group. The stereoacuity levels may have been related to the duration of alignment before the constant esotropia developed, but it was not studied because this information was not always available. However, the clinical relevance of differences reported may be considered low, because differences in deviation are in the limit of reproducibility of prism and alternate cover test.[9]

Notwithstanding, the reported percentage of success, as defined in the present study, was higher in the botulinum group at middle term. Apparently, there was a slight trend toward increasing deviation in the surgery group. Loss of effect with time could be explained by the technique we used for posterior fixation, i.e., the classical technique of muscle scleral fixation, but done immediately anterior to the level of the muscle pulley, which is located in a coronal plane just behind the equator of the globe.[10] Posterior fixation in our cases was done at approximately 13–14 mm from the insertion, and we used nonabsorbable suture to minimize undercorrection. We had no overcorrections in either study group. In the past while performing 0.5 to 1 mm larger recessions combined with posterior fixation we had overcorrections. These frequently increased with time and required reoperation. In the present study we thus adhered strictly to the AAO guidelines.[8]

A limitation of the present investigation is that data were collected retrospectively, so there is potential bias in treatment assignment and patient selection. However, differences were not observed in pretreatment characteristics between the two groups, except for refraction. A difference of 0.25 D in spherical equivalent is not considered of clinical significance.

In this study, the odds in favor of successful result was greater in the botulinum group, and the odds in favor of using bifocal, in the surgery group. But reintervention was not indicated with significantly greater frequency in the surgery compared with the botulinum group. Good apparent aesthetic result and detectable binocularity were factors in favor of discarding reintervention in limit cases (with distance deviation of approximately 10 PD), which could contribute to similar frequency of reintervention.

Interestingly, not only treatment modality and initial deviation were significant predictor variables on deviation after treatment and success, but also lines of difference in visual acuity between the two eyes, particularly at one year of follow-up. In theory, amblyopia could have influence on the result of surgery, recurrence of deviation, distance-near disparity, and binocular vision.

We believe that the apparent superiority of botulinum toxin at middle term, could be due to unnoticed bias, and specific surgical technique. A justification for more conservative (i.e., less posterior) scleral fixation of the medial rectus muscles is the risk of overcorrection with time that experts have frequently described in association with fadenoperation, in particular when combined with medial rectus recession.[1,3] According to our previous experience,

increasing the amount of medial rectus recessions involves greater risk of overcorrection with time at distance, although motor success results could improve at short term. The results at near could be better with a more posterior placement of posterior fixation. Although in theory there is no significant effect of this procedure on deviation at distance, fadenoperation has been mentioned as a contributing factor to distance overcorrection when combined with muscle recession. However, as mentioned above, more recent studies demonstrate that the effect of posterior fixation is stable at long term.[11]

## Conclusions

To summarize, we found that botulinum toxin could be superior to, or at least as effective as, surgery, at middle term, in the treatment of partially accommodative esotropia with high AC/A ratio. Although this is a relatively surprising finding, botulinum toxin appears as a useful therapeutic alternative in this type of strabismus, with remarkable results in binocularity. A randomized controlled trial is necessary to confirm these results, and extend conclusions to common clinical practice.

## Supporting information

**S1 File.**
(SAV)

## Author Contributions

**Conceptualization:** Jaime Tejedor, Francisco J. Gutiérrez-Carmona.

**Formal analysis:** Jaime Tejedor, Francisco J. Gutiérrez-Carmona.

**Funding acquisition:** Jaime Tejedor.

**Investigation:** Jaime Tejedor.

**Methodology:** Jaime Tejedor.

**Project administration:** Jaime Tejedor.

**Supervision:** Francisco J. Gutiérrez-Carmona.

**Validation:** Francisco J. Gutiérrez-Carmona.

**Writing – original draft:** Jaime Tejedor, Francisco J. Gutiérrez-Carmona.

**Writing – review & editing:** Jaime Tejedor, Francisco J. Gutiérrez-Carmona.

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
