## [Decision Letter · Decision Letter 0]

11 Dec 2019

PONE-D-19-32474

Botulinum toxin in the treatment of high AC/A ratio accommodative esotropia

PLOS ONE

Dear Dr. Tejedor,

Thank you for submitting your manuscript to PLOS ONE. After careful consideration, we feel that it has merit but does not fully meet PLOS ONE’s publication criteria as it currently stands. Therefore, we invite you to submit a revised version of the manuscript that addresses the points raised during the review process.

We would appreciate receiving your revised manuscript by Jan 24 2020 11:59PM. To enhance the reproducibility of your results, we recommend that if applicable you deposit your laboratory protocols in protocols.io, where a protocol can be assigned its own identifier (DOI) such that it can be cited independently in the future. For instructions see: http://journals.plos.org/plosone/s/submission-guidelines#loc-laboratory-protocols

We look forward to receiving your revised manuscript.

Kind regards,

Ahmed Awadein, MD, Ph.D, FRCS

Academic Editor

PLOS ONE

Journal Requirements:

**When submitting your revision, we need you to address these additional requirements:**

**Please ensure that your manuscript meets PLOS ONE's style requirements, including those for file naming. The PLOS ONE style templates can be found at http://www.plosone.org/attachments/PLOSOne_formatting_sample_main_body.pdf and http://www.plosone.org/attachments/PLOSOne_formatting_sample_title_authors_affiliations.pdf****Thank you for including your ethics statement: The study was approved by the Institutional Ethics Committee (MINECO UAMA13-4E-2192) and adhered to the tenets of the Declaration of Helsinki**

Reviewers' comments:

Reviewer's Responses to Questions

**Comments to the Author**

1. Is the manuscript technically sound, and do the data support the conclusions?

Reviewer #1: Partly

Reviewer #2: Yes

2. Has the statistical analysis been performed appropriately and rigorously? 

Reviewer #1: Yes

Reviewer #2: Yes

3. Have the authors made all data underlying the findings in their manuscript fully available?

Reviewer #1: Yes

Reviewer #2: Yes

4. Is the manuscript presented in an intelligible fashion and written in standard English?

Reviewer #1: No

Reviewer #2: No

5. Review Comments to the Author

Reviewer #1: Re: Manuscript PONE-D-19-32474 "Botulinum toxin in the treatment of high AC/A ratio accommodative esotropia".

Dependant on the author’s response to some of the questions I have posed,Irecommend that the article be accepted with Minor(?) revision,English is not the authors mother tongue, and the language needs to be corrected in many areas to diminish ambiguity. (While nevertheless appreciating that it is not easy to write in a language different to one’s home language)

The study is retrospective and comparative, and includes patients with accommodative esotropia (ET) [Because of the residual deviation despite wearing the full cycloplegic correction,this should be changed to partially accommodative ET{PAET} with a high accommodative convergence/accommodation ratio (AC/A).]The patients were divided into two groups. Group one patients were treated by botulinum toxin injection to both medial rectus muscles. Group two patients were treated by bilateral medial rectus muscle recessions, augmented with posterior fixation. The motor and sensory outcomes of patients in each group were compared.

While the manuscript is technically sound in most respects, I am concerned about the following: Patients received glasses based on the cycloplegic refraction at the initial visit, which were then worn for two months. The ET angle was measured at the next visit (the baseline visit). At that visit patients who had a residual distance ET of at least 10 prism dioptres (PD), and an ET at near that was larger than the distance ET by 10 PD or more were considered eligible for inclusion in Group 1 or 2(Lines 108-112).

It is considered by many that if the glasses do not reduce the distance deviation to less than 10 PD, another cycloplegic refraction, which usually discloses additional hyperopic refractive error, should be performed. If this is found, the spectacle power should be increased, and the glasses worn for at least another 4 weeks. If the new spectacles reduce the ET angle the refraction should again be repeated. This process may reduce or eliminate the distance deviation ,and reduce the distance/ near disparity.

From the manuscript it appears as if this was not done.(Lines 51and 52-“Cycloplegic refraction was carried out at the initial visit, at 6 months and a year after BTX injection or surgery”) Could the authors please comment?

Line 82 -88

Consider changing to “ …….with additional procedures, several of which are available………….deviation at near. We had used the later procedure for several years,………

for this condition.”

Line 85

Throughout the whole manuscript, change “medial recti and medial recti muscles” to “medial rectus muscles”, and change “bimedial recession” to “bilateral medial rectus muscle recessions”.

Line90.

“… parents did not agree to surgery. The results ………………..were considered to be satisfactory.”

Line104 .

Suggest change to “After a detailed explanation of the nature of the study, informed consent was obtained for BTX injection or surgery, and for collection of the relevant data……………”

Line 125.

Children with “………….myopia greater than -0.50D were excluded.” Were any myopic children include in the study? ( Line 174 “or myopic whose visual acuity improved with correction” )[Table 1 appears to indicate all patients were greater than +2.25 D hyperopic])

Line 133 .

Cycloplegic retinoscopy ………were also part of the study.(but only “at the initial visit, at 6 months and a year after BTX injection or surgery”) ?

Line 148.

Suggest change to: “…of the muscle pulley. Although this was identified during surgery,we nevertheless chose to use a classical scleral fixation technique to secure the muscle, with a non-absorbable 5-0 Dacron suture, and not a modified pulley fixation technique, as described by Clarke et al5”(published in 2004 ,and not later).

Line 158 -160

This is repetitive of lines 134-136

Lines 187 -192

Suggest change to “Six months after Botox injection or surgery, the deviation (with distance glasses) at distance and near, as well as stereoacuity were similar in the two groups. However, the deviations in the surgery group were significantly larger and stereoacuity in the BTX group significantly better at I year.”

Lines 191-192 states “stereoacuity was significantly better in the surgery group at I year”, while lines 238-239 and 350 (the legend to figure 2) state it was “better in the BTX group”. Please correct

The stereoacuity levels may have been related the duration of alignment before the constant ET developed(not studied)

Line 202-208.

“The percentage of success ….was similar in both groups……. at 6 months, but smaller in the surgery group at I year……...”

Table 1 contains the preintervention median, minimum and maximum refractions. Line 52 indicates that cycloplegic refractions were carried at the initial visit and 6 and 12 months after BTX or surgery. The manuscript does not report if any of the patients required a stronger hyperopic correction between the 6 and 12 month post- treatment visits, or in those requiring a new procedure (line 214). Any under correction could have influenced the deviations, stereoacuity, distance near disparity and the need for bifocals. As the authors point out (lines 193-196), whether statistically significant or not, most differences were of little clinical relevance

Line 255.

Replace with “We had no overcorrections in either study group. In the past, while performing larger recessions of x mm (authors to add in size of recessions), combined with posterior fixation we had overcorrections. These frequently increased with time and required reoperation. In the present study we thus adhered strictly to the AAO guidelines 6 ”

Line 279-284.

Could the authors comment on whether they could either slightly increase the amount of medial rectus muscle recession, or alter placement of the PFS, in order to improve the result of surgery with PFS without a greater risk of overcorrection

Line 280

By “more conservative(i.e.,less posterior)scleral……..” Do they mean placing the PFS more anteriorly, which should have less effect on the near deviation, but not on the distance deviation .If so how would it reduce the overcorrection rate?

955 Words

Reviewer #2: The manuscript describes a novel way for treating partially-accommodative esotropia with botulinum toxin

1- Overall the manuscript needs extensive grammatical revision

2- Throughout the manuscript, consider replacing high accommodative convergence/accommodation (AC/A) ratio accommodative esotropia with partially acommodiative esotropia with high AC/A ratio

3- Can the authors clarify the relatively young group of the included patients? Why were older age groups not included?

4- Lines 101-102: What standard deviation was used to allow sample size calculation?

5- Did the authors include patients with convergence excess ET with normal AC/A ratio? and if not why were they included, though they might benefit from the same procedure.

6- Line 146 The relatively small amount of recession implies that the authors used standard recession rather than augmented recession. Usually the amounts of recession would be much higher for augmented recession

7- Line 174: Didnt you exclude myopic patients?

8- Lines 248-255 the authors are referring to pulley fixation rather than posterior scleral fixation. This section is not related to the current work and can be trimmed

6. PLOS authors have the option to publish the peer review history of their article (what does this mean?). If published, this will include your full peer review and any attached files.

Reviewer #1: No

Reviewer #2: No

---

## [Author Response · Author response to Decision Letter 0]

6 Jan 2020

We list changes made in response to the reviewers and editorial comments. Page numbers are those of the clean revised version:

1. We checked the Plos One style requirements

2. We included the specific name of the Ethics Committee in the Methods section and submission system ‘Ethics statement’, line 106

3. We ensured that each author is linked to an affiliation, lines 15-20

Reviewers comments

Reviewer 1

We changed title to ‘Botulinum toxin in the treatment of partially accommodative esotropia with high AC/A ratio’, lines 7-8.

Due to space limits and to avoid misunderstanding we summarized the initial evaluation and baseline visit as ‘initial visit’ in the abstract section (lines 51-53 in earlier version). Only when we were sure that children wore full plus glasses full time for 2 months, the indication of botulinum toxin injection or surgery was made, when deviation at distance was 10 PD or more. Children were refracted in each visit, so they were re-refracted at 2 months to ensure additional plus was not required. We checked in our records that 3 patients in the botulinum group and 2 patients in the surgery group required two more months of wearing glasses full time with full plus power, before a decision on new intervention was taken. We have clarified this in the abstract (line 53) methods and methods section (lines 112-116) of the manuscript.

Line 82-88: change made (lines 83-90).

Line 85: change done throughout the manuscript.

Line 90: change done (lines 91-93).

Line 104: done (lines 108-109).

Line 125: Three children in the study search were myopic but were excluded. The comment in line 174 (a general comment on refraction of myopes) has been deleted to avoid the interpretation that myopic children were participants.

Line 133: cycloplegic retinoscopy… were part of the study in each follow-up visit. It has been specified now in the methods section (lines 141-142).

Line 148: change done (lines 157-161)

Lines 158-160: deleted

Lines 187-192: we have rewritten the paragraph and clarified that stereoacuity was better in the Botox group at 1 year (lines 195-199). We added a sentence in the discussion section to point out that the stereoacuity levels may have been related to the duration of alignment before the constant esotropia developed, but it was not studied because this information was not always available (lines 252-255).

Lines 202-208: we checked how many patients required change in refraction after cycloplegic retinoscopy at 6 and 12 months and in those requiring a new procedure, and have added this information in the results section. 4 patients in the botulinum group and 3 patients in the surgery group required change (additional plus) in refraction at 6 months, whereas 5 patients in the botulinum and 4 in the surgery group required new glasses prescription at one year (lines 217-220). One of the children who required a new procedure in the botulinum group and 2 in the surgery group were prescribed additional hyperopic correction before indication of reintervention (lines 227-229).

Line 255: change done (lines 266-270).

Lines 279-284, line 280: we have added several comments at the end of the discussion (lines 290-301).

Reviewer 2

1. Grammatical revision was done according to the reviewers and editorial indications

2. We replaced high accommodative convergence/accommodation ratio accommodative esotropia with partially accommodative esotropia with high AC/A ratio throughout the manuscript.

3. We did not exclude older age children. We have included children at the age of diagnosis and treatment. The typical onset of accommodative esotropia is between 6 months and 7 years (average 2.5 y) (AAO Basic and Clinical Science Course 2018) and the range of age in our cohort was 3 to 8 years old. Apparently, children with high AC/A ratio have a younger age at presentation (Parks MM, Arch Ophthalmol 1958; 59:364-80).

4. Lines 101-102: We used 12 PD as standard deviation for calculation, based on initial collected data, and so that the size of the sample was big enough. We added this information in the methods section

5. We have used botulinum toxin in convergence excess with normal high AC/A ratio but didn’t include these patients in the present study, numbers are very small yet.

6. Line 146: We used standard recessions to minimize the risk of overcorrection when posterior fixation is added to an augmented recession. We included comment at the end of the discussion (last paragraph).

7. Line 174: Three children in the study search were myopic but were excluded. The comment in line 174 (a general comment on refraction of myopes) has been deleted to avoid the interpretation that myopic children were participants.

8. Lines 248-255: we deleted comments on pulley fixation.

---

## [Editor Report · Decision Letter 1]

4 Feb 2020

Botulinum toxin in the treatment of partially accommodative esotropia with high AC/A ratio

PONE-D-19-32474R1

Dear Dr. Tejedor,

We are pleased to inform you that your manuscript has been judged scientifically suitable for publication and will be formally accepted for publication once it complies with all outstanding technical requirements.

With kind regards,

Ahmed Awadein, MD, Ph.D, FRCS

Academic Editor

PLOS ONE
---

## [Editor Report · Acceptance letter]

13 Feb 2020

PONE-D-19-32474R1 

Botulinum toxin in the treatment of partially accommodative esotropia with high AC/A ratio 

Dear Dr. Tejedor:

I am pleased to inform you that your manuscript has been deemed suitable for publication in PLOS ONE. Congratulations! Your manuscript is now with our production department. 

With kind regards,

on behalf of

Dr. Ahmed Awadein 

Academic Editor

PLOS ONE